# Multi-objective training of Generative Adversarial Networks with multiple discriminators

## Abstract

Recent literature has demonstrated promising results on the training of Generative Adversarial Networks by employing a set of discriminators, as opposed to the traditional game involving one generator against a single adversary. Those methods perform single-objective optimization on some simple consolidation of the losses, e.g. an average. In this work, we revisit the multiple-discriminator approach by framing the simultaneous minimization of losses provided by different models as a multi-objective optimization problem. Specifically, we evaluate the performance of multiple gradient descent and the hypervolume maximization algorithm on a number of different datasets. Moreover, we argue that the previously proposed methods and hypervolume maximization can all be seen as variations of multiple gradient descent in which the update direction computation can be done efficiently. Our results indicate that hypervolume maximization presents a better compromise between sample quality and diversity, and computational cost than previous methods.

## 1 Introduction

Generative Adversarial Networks (GANs) (Goodfellow et al., 2014) offer a new approach to generative modeling, using game-theoretic training schemes to implicitly learn a given probability density. Prior to the emergence of GAN architectures, realistic generative modeling remained elusive. When offering unparalleled realism, GAN training remains fraught with stability issues. Commonly reported shortcomings involved in the GAN game are the lack of useful gradients provided by the discriminator, and mode collapse, i.e. lack of diversity in the generator's samples.

Considerable research effort has been devoted in recent literature in order to overcome training instability [1] within the GAN framework. Some architectures such as BEGAN (Berthelot et al., 2017) have applied auto-encoders as discriminators and proposed a new loss to help stabilize training. Methods such as TTUR (Heusel et al., 2017), in turn, have attempted to define schedules for updating the generator and discriminator differently. The PacGAN algorithm (Lin et al., 2017) proposes to modify the discriminator's architecture which will receive $m$ concatenated samples as input, while modifications to alternate updates in SGD were introduced in (Yadav et al., 2017). These samples are jointly classified as either real or generated, and authors show that this enforces sample diversity. In SNGAN (Miyato et al., 2018), authors introduce spectral normalization on the discriminator aiming to ensure Lipschitz continuity, which is empirically shown to consistently yield high quality samples when different sets of hyperparameters are used.

Recent works have proposed to tackle GANs instability issues using multiple discriminators. Neyshabur et al. (2017) propose a GAN variation in which one generator is trained against a set of discriminators, where each discriminator sees a fixed random projection of the inputs. Prior work, including GMAN (Durugkar et al., 2016) has also explored training against multiple discriminators.

---

[1] Instability in the sense commonly used in GANs literature, i.e. divergence and mode-collapse of the generator when the discriminator is able to easily distinguish real and fake samples during training (Neyshabur et al., 2017; Arjovsky et al., 2017; Berthelot et al., 2017).

In this paper, we build upon Neyshabur et al.'s introduced framework and propose reformulating the average loss minimization aiming to further stabilize GAN training. Specifically, we propose treating the loss signal provided by each discriminator as an independent objective function. To achieve this, we simultaneously minimize the losses using multi-objective optimization techniques. Namely, we exploit previously introduced methods in literature such as the multiple gradient descent algorithm (MGD) (Désidéri, 2012). However, due to MGD's prohibitively high cost in the case of large neural networks, we propose the use of more efficient alternatives such as maximization of the hypervolume of the region defined between a fixed, shared upper bound on those losses, which we will refer to as the *nadir point* $\boldsymbol{\eta}^*$, and each of the component losses.

In contrast to Neyshabur et al. (2017)'s approach, where the average loss is minimized when training the generator, hypervolume maximization (HV) optimizes a weighted loss, and the generator's training will adaptively assign greater importance to feedback from discriminators against which it performs poorly.

Experiments performed on MNIST show that HV presents a good compromise in the *computational cost-samples quality* trade-off, when compared to average loss minimization or GMAN's approach (low quality and cost), and MGD (high quality and cost). Also, the sensitivity to introduced hyperparameters is studied and results indicate that increasing the number of discriminators consequently increases the generator's robustness along with sample quality and diversity. Experiments on CIFAR-10 indicate the method described produces higher quality generator samples in terms of quantitative evaluation. Moreover, image quality and sample diversity are once more shown to consistently improve as we increase the number of discriminators.

In summary, our main contributions are the following:

1. We offer a new perspective on multiple-discriminator GAN training by framing it in the context of multi-objective optimization, and draw similarities between previous research in GANs variations and MGD, commonly employed as a general solver for multi-objective optimization.

2. We propose a new method for training multiple-discriminator GANs: Hypervolume maximization, which weighs the gradient contributions of each discriminator by its loss.

The remainder of this document is organized as follows: Section 2 introduces definitions on multi-objective optimization and MGD. In Section 3 we describe prior relevant literature. Hypervolume maximization is detailed in Section 4, with experiments and results presented in Section 5. Conclusions and directions for future work are drawn in Section 6.

## 2 Preliminaries

In this section we provide some definitions regarding multi-objective optimization literature which will be useful in the next sections. Henceforth, the boldface notation will be used to indicate vector-valued variables.

**Multi-objective optimization.** A multi-objective optimization problem is defined as (Deb, 2001):

$$\min \mathbf{F}(\mathbf{x}) = [f_1(\mathbf{x}), f_2(\mathbf{x}), ..., f_K(\mathbf{x})]^T,$$
$$\mathbf{x} \in \Omega, \tag{1}$$

where $K$ is the number of objectives, $\Omega$ is the variables space and $\mathbf{x} = [x_1, x_2, ..., x_n]^T \in \Omega$ is a decision vector or possible solution to the problem. $\mathbf{F} : \Omega \to \mathbb{R}^K$ is a set of $K$-objective functions that maps the $n$-dimensional variables space to the $K$-dimensional objective space.

**Pareto-dominance.** Let $\mathbf{x}_1$ and $\mathbf{x}_2$ be two decision vectors. $\mathbf{x}_1$ is said to dominate $\mathbf{x}_2$ (denoted by $\mathbf{x}_1 \prec \mathbf{x}_2$) if and only if $f_i(\mathbf{x}_1) \leq f_i(\mathbf{x}_2)$ for all $i \in \{1, 2, \ldots, K\}$ and $f_j(\mathbf{x}_1) < f_j(\mathbf{x}_2)$ for some $j \in \{1, 2, \ldots, K\}$. If a decision vector $\mathbf{x}$ is dominated by no other vector in $\Omega$, $\mathbf{x}$ is said to be non-dominated.

**Pareto-optimality.** A decision vector $\mathbf{x}^* \in \Omega$ is said to be Pareto-optimal if and only if there is no $\mathbf{x} \in \Omega$ such that $\mathbf{x} \prec \mathbf{x}^*$, i.e. $\mathbf{x}^*$ is a non-dominated solution. The Pareto-optimal Set (PS) is

defined as the set of all Pareto-optimal solutions $\mathbf{x} \in \Omega$, i.e., $PS = \{\mathbf{x} \in \Omega | \mathbf{x}$ is Pareto optimal$\}$. The set of all objective vectors $\mathbf{F}(\mathbf{x})$ such that $\mathbf{x}$ is Pareto-optimal is called Pareto front (PF), that is $PF = \{\mathbf{F}(\mathbf{x}) \in \mathbb{R}^K | \mathbf{x} \in PS\}$.

**Pareto-stationarity.** Pareto-stationarity is a necessary condition for Pareto-optimality. For $f_k$ differentiable everywhere for all $k$, $\mathbf{F}$ is said to be Pareto-stationary at the point $\mathbf{x}$ if there exists a set of scalars $\alpha_k, k \in \{1, \dots, K\}$, such that:

$$\sum_{k=1}^{K} \alpha_k \nabla f_k = \mathbf{0}, \quad \sum_{k=1}^{K} \alpha_k = 1, \quad \alpha_k \geq 0 \quad \forall k. \tag{2}$$

**Multiple Gradient Descent.** Multiple gradient descent (Désidéri, 2012; Schäffler et al., 2002; Peitz & Dellnitz, 2018) was proposed for the unconstrained case of multi-objective optimization of $\mathbf{F}(\mathbf{x})$ assuming a convex, continuously differentiable and smooth $f_k(\mathbf{x})$ for all $k$. MGD finds a common descent direction for all $f_k$ by defining the convex hull of all $\nabla f_k(\mathbf{x})$ and finding the minimum norm element within it. Consider $\mathbf{w}^*$ given by:

$$\mathbf{w}^* = \operatorname{argmin}||\mathbf{w}||, \quad \mathbf{w} = \sum_{k=1}^{K} \alpha_k \nabla f_k(\mathbf{x}), \quad \text{s.t.} \quad \sum_{k=1}^{K} \alpha_k = 1, \quad \alpha_k \geq 0 \quad \forall k. \tag{3}$$

$\mathbf{w}^*$ will be either $\mathbf{0}$ in which case $\mathbf{x}$ is a Pareto-stationary point, or $\mathbf{w}^* \neq \mathbf{0}$ and then $\mathbf{w}^*$ is a descent direction for all $f_i(\mathbf{x})$. Similar to gradient descent, MGD consists in finding the *common* steepest descent direction $\mathbf{w}_t^*$ at each iteration $t$, and then updating parameters with a learning rate $\lambda$ according to $\mathbf{x}_{t+1} = \mathbf{x}_t - \lambda \frac{\mathbf{w}_t^*}{||\mathbf{w}_t^*||}$.

# 3 RELATED WORK

## 3.1 TRAINING GANs WITH MULTIPLE DISCRIMINATORS

While we would prefer to always have strong gradients from the discriminator during training, the vanilla GAN makes this difficult to ensure, as the discriminator quickly learns to distinguish real and generated samples (Goodfellow, 2016), thus providing no meaningful error signal to improve the generator thereafter. Durugkar et al. (2016) proposed the Generative Multi-Adversarial Networks (GMAN) which consist in training the generator against a *softmax* weighted arithmetic average of $K$ different discriminators, according to Eq. 4.

$$\mathcal{L}_G = \sum_{k=1}^{K} \alpha_k \mathcal{L}_{D_k}, \tag{4}$$

where $\alpha_k = \frac{e^{\beta \mathcal{L}_{D_k}}}{\sum_{j=1}^{K} e^{\beta \mathcal{L}_{D_j}}}$, $\beta \geq 0$, and $\mathcal{L}_{D_k}$ is the loss of discriminator $k$ and defined as

$$\mathcal{L}_{D_k} = -\mathbb{E}_{\mathbf{x} \sim p_{\text{data}}} \log D_k(\mathbf{x}) - \mathbb{E}_{\mathbf{z} \sim p_z} \log(1 - D_k(G(\mathbf{z}))), \tag{5}$$

where $D_k(\mathbf{x})$ and $G(\mathbf{z})$ are the outputs of the $k$-th discriminator and the generator, respectively. The goal of using the proposed averaging scheme is to privilege worse discriminators and thus providing more useful gradients to the generator during training. Experiments were performed with $\beta = 0$ (equal weights), $\beta \to \infty$ (only worst discriminator is taken into account), $\beta = 1$, and $\beta$ learned by the generator. Models with $K = \{2, 5\}$ were tested and evaluated using a proposed metric and the Inception score (Salimans et al., 2016). However, results showed that the simple average of discriminator's losses provided the best values for both metrics in most of the considered cases.

Opposed to GMAN, Neyshabur et al. (2017) proposed training a GAN with $K$ discriminators using the same architecture. Each discriminator $D_k$ sees a different randomly projected lower-dimensional version of the input image. Random projections are defined by a randomly initialized matrix $W_k$, which remains fixed during training. Theoretical results provided show that the distribution induced by the generator $G$ will converge to the real data distribution $p_{\text{data}}$, as long as there is a sufficient number of discriminators. Moreover, discriminative tasks in the projected space are harder, i.e. real

and fake samples are more alike, thus avoiding early convergence of discriminators, which leads to common stability issues in GAN training such as mode-collapse (Goodfellow, 2016). Essentially, the authors trade one hard problem for $K$ easier subproblems. The losses of each discriminator $\mathcal{L}_{D_k}$ are the same as shown in Eq. 5. However, the generator loss $\mathcal{L}_G$ is defined as simply the sum of the losses provided by each discriminator, as shown in Eq. 6. This choice of $\mathcal{L}_G$ does not exploit available information such as the performance of the generator with respect to each discriminator.

$$\mathcal{L}_G = -\sum_{k=1}^{K} \mathbb{E}_{\mathbf{z} \sim p_z} \log D_k(G(\mathbf{z})). \tag{6}$$

### 3.2 HYPERVOLUME MAXIMIZATION

Consider a set of solutions $S$ for a multi-objective optimization problem. The hypervolume $\mathcal{H}$ of $S$ is defined as (Fleischer, 2003): $\mathcal{H}(S) = \mu(\cup_{\mathbf{x} \in S} [\mathbf{F}(\mathbf{x}), \boldsymbol{\eta}^*])$, where $\mu$ is the Lebesgue measure and $\boldsymbol{\eta}^*$ is a point dominated by all $\mathbf{x} \in S$ (i.e. $f_i(\mathbf{x})$ is upper-bounded by $\eta$), referred to as *nadir point*. $\mathcal{H}(S)$ can be understood as the size of the space covered by $\{\mathbf{F}(\mathbf{x}) | \mathbf{x} \in S\}$ (Bader & Zitzler, 2011).

The hypervolume was originally introduced as a quantitative metric for coverage and convergence of Pareto-optimal fronts obtained through population based algorithms (Beume et al., 2007). Methods based on direct maximization of $\mathcal{H}$ exhibit favorable convergence even in challenging scenarios, such as simultaneous minimization of 50 objectives (Bader & Zitzler, 2011). In the context of Machine Learning, a single-solution hypervolume maximization has been applied to neural networks as a surrogate loss for mean squared error (Miranda & Zuben, 2016), i.e. the loss provided by each example in a training batch is treated as a single cost and the multi-objective approach aims to minimize costs over all examples. Authors show that such method provides an inexpensive boosting-like training.

## 4 MULTI-OBJECTIVE TRAINING OF GANS WITH MULTIPLE DISCRIMINATORS

We introduce a variation of the GAN game such that the generator solves the following multi-objective problem:

$$\min \boldsymbol{\mathcal{L}}_G(\mathbf{x}) = [l_1(\mathbf{z}), l_2(\mathbf{z}), ..., l_K(\mathbf{z})]^T, \tag{7}$$

where each $l_k = -\mathbb{E}_{z \sim p_z} \log D_k(G(z))$, $k \in \{1, ..., K\}$, is the loss provided by the $k$-th discriminator. Training proceeds as the usual formulation (Goodfellow et al., 2014), i.e. with alternate updates between the discriminators and the generator. Updates of each discriminator are performed to minimize the loss described in Eq. 5.

A natural choice for generator's updates is the MGD algorithm, described in Section 2. However, computing the direction of steepest descent $\mathbf{w}^*$ before every parameter update step, as required in MGD, can be prohibitively expensive for large neural networks. Therefore, we propose an alternative scheme for multi-objective optimization and argue that both our proposal and previously published methods can all be viewed as performing computationally more efficient versions of MGD update rule without the burden of having to solve a quadratric program, i.e. computing $\mathbf{w}^*$, every iteration.

### 4.1 HYPERVOLUME MAXIMIZATION FOR TRAINING GANS

Fleischer (Fleischer, 2003) has shown that maximizing $\mathcal{H}$ yields Pareto-optimal solutions. Since MGD converges to a set of Pareto-stationary points, i.e. a super-set of the Pareto-optimal solutions, hypervolume maximization yields a sub-set of the solutions obtained using MGD.

We exploit the above mentioned property and define the generator loss as the negative log-hypervolume, as defined in Eq. 8:

$$\mathcal{L}_G = -\mathcal{V} = -\sum_{k=1}^{K} \log(\eta - l_k), \tag{8}$$

where the nadir point coordinate $\eta$ is an upper bound for all $l_k$. In Fig. 1 we provide an illustrative example for the case where $K = 2$. The highlighted region corresponds to $e^{\mathcal{V}}$. Since the nadir point $\boldsymbol{\eta}^*$ is fixed, $\mathcal{V}$ will only be maximized, and consequently $\mathcal{L}_G$ minimized, if each $l_k$ is minimized.

Moreover, by adapting the results shown in (Miranda & Zuben, 2016), the gradient of $\mathcal{L}_G$ with respect to any generator's parameter $\theta$ is given by:

$$\frac{\partial \mathcal{L}_G}{\partial \theta} = \sum_{k=1}^{K} \frac{1}{\eta - l_k} \frac{\partial l_k}{\partial \theta}. \qquad (9)$$

In other words, the gradient can be obtained by computing a weighted sum of the gradients of the losses provided by each discriminator, whose weights are defined as the inverse distance to the nadir point components. This formulation will naturally assign more importance to higher losses in the final gradient, which is another useful property of hypervolume maximization.

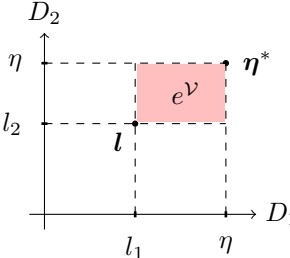

Figure 1: 2D example of the objective space where the generator loss is being optimized.

**Nadir point selection.** It is evident from Eq. 9 that the selection of $\eta$ directly affects the importance assignment of gradients provided by different discriminators. Particularly, as the quantity $\min_k\{\eta - l_k\}$ grows, the multi-objective GAN game approaches the one defined by the simple average of $l_k$. Previous literature has discussed in depth the effects of the selection of $\eta$ in the case of population-based methods (Auger et al., 2009; 2012). However, those results are not readily applicable for the single-solution case. As will be shown in Section 5, our experiments indicate that the choice of $\eta$ plays an important role in the final quality of samples. Nevertheless, this effect becomes less relevant as the number of discriminators increases.

**Nadir point adaptation.** Similarly to (Miranda & Zuben, 2016), we propose an adaptive scheme for $\eta$ such that at iteration $t$: $\eta_t = \delta \max_k\{l_{k,t}\}$, where $\delta > 1$ is a user-defined parameter which will be referred to as *slack*. This enforces $\min_k\{\eta - l_k\}$ to be higher when $\max_k\{l_{k,t}\}$ is high and low otherwise, which induces a similar behavior as an average loss when training begins and automatically places more importance on the discriminators in which performance is worse as training progresses. Extra discussion and an illustrative example of the adaptation scheme adopted is presented in Appendix G.

**Comparison to average loss minimization.** The upper bound proven by Neyshabur et al. (2017) assumes that the marginals of the real and generated distributions are identical along all random projections. Average loss minimization does not ensure equally good approximation between the marginals along all directions. In case of a trade-off between discriminators, i.e. if decreasing the loss on a given projection increases the loss with respect to another one, the distribution of losses can be uneven. With HV on the other hand, especially when $\eta$ is reduced throughout training, overall loss will be kept high as long as there are discriminators with high loss. This objective tends to prefer central regions of a trade-off, in which all discriminators present a roughly equally low loss.

## 4.2 Relationship between multiple discriminator GANs and MGD

All methods described previously for the solution of GANs with multiple discriminators, i.e. average loss minimization (Neyshabur et al., 2017), GMAN's weighted average (Durugkar et al., 2016) and hypervolume maximization can be defined as MGD-like two-step algorithms consisting of: *Step 1 -* consolidating all gradients into a single update direction (compute the set $\alpha_{1,...,K}$); *Step 2 -* updating parameters in the direction returned in step 1. Definition of *Step 1* for the different methods studied here can be seen in the following:

1. MGD: $\alpha_{1:K} = \operatorname{argmin}_\alpha ||\mathbf{w}||$,    s.t.    $\sum_{k=1}^{K} \alpha_k = 1$,    $\alpha_k \geq 0 \; \forall k \in \{1, ..., K\}$

2. Average loss minimization (Neyshabur et al., 2017): $\alpha_k = \frac{1}{K}$

3. GMAN (Durugkar et al., 2016): $\alpha_k = \operatorname{softmax}(l_{1:K})_k$

4. Hypervolume maximization: $\alpha_k = \frac{1}{T(\eta - l_k)}$,    $T = \sum_{k=1}^{K} \frac{1}{\eta - l_k}$

## 5 EXPERIMENTS

We performed three sets of experiments aiming to analyze the following aspects: (i) How alternative methods for training GANs with multiple discriminators perform in comparison to MGD; (ii) How alternative methods perform in comparison to each other in terms of sample quality and coverage; and (iii) Whether the behavior induced by HV improves the results with respect to the baseline methods.

Firstly, we exploited the relatively low dimensionality of MNIST and used it as testbed for a comparison of MGD with the other approaches, i.e. average loss minimization (AVG), GMAN's weighted average loss, and HV, proposed in this work. Moreover, multiple initializations and *slack* combinations were evaluated in order to investigate how varying the number of discriminators affects robustness to those factors.

Then, experiments were performed with CIFAR-10 while increasing the number of discriminators. We evaluated HV's performance compared to baseline methods, and the effect in samples quality. We also analyzed the impact on the diversity of generated samples by using the stacked MNIST dataset (Srivastava et al., 2017). Samples of generators trained on stacked MNIST, CIFAR-10, CelebA, and Cats dataset are shown in the Appendix.

In all experiments performed, the same architecture, set of hyperparameters and initialization were used for both AVG, GMAN and our proposed method. The only different aspect is the generator loss. Unless stated otherwise, Adam (Kingma & Ba, 2014) was used to train all the models with learning rate, $\beta_1$ and $\beta_2$ set to 0.0002, 0.5 and 0.999, respectively. Mini-batch size was set to 64. The Fréchet Inception Distance (FID) (Heusel et al., 2017) was employed for comparison. Details on FID computation can be found in Appendix A.

### 5.1 MGD COMPARED WITH ALTERNATIVE METHODS

We employed MGD in our experiments with MNIST. In order to do so, a quadratic program has to be solved prior to every parameters update. For this, we used the Scipy's implementation of the Serial Least Square Quadratic Program solver[2].

Three and four fully connected layers with *LeakyReLU* activations were used for the generator and discriminator, respectively. Dropout was also employed in the discriminator and the random projection layer was implemented as a randomly initialized norm-1 fully connected layer, reducing the vectorized dimensionality of MNIST from 784 to 512. A pretrained *LeNet* (LeCun et al., 1998) was used for FID computation.

Experiments over 100 epochs with 8 discriminators are reported in Fig. 2 and Fig. 3. In Fig. 2, box-plots refer to 30 independent computations of FID over 10000 images sampled from the generator which achieved the minimum FID at train time. FID results are measured at train time over 1000 images and the best values are reported in Fig. 3 along with the necessary time to achieve it.

MGD outperforms all tested methods. However, its cost per iteration does not allow its use in more relevant datasets other than MNIST. Hypervolume maximization, on the other hand, performs closest to MGD than the considered baselines, while introducing no relevant extra cost. In Fig. 4, we analyze convergence in the Pareto-stationarity sense by plotting the norm of the update direction for each method, given by $||\sum_{k=1}^{K} \alpha_k \nabla l_k||$. All methods converged to similar norms, leading to the conclusion that different Pareto-stationary solutions will perform differently in terms of quality of samples. FID as a function of wall-clock time is shown in Figure 22 (Appendix H).

**HV sensitivity to initialization and choice of** $\delta$. Analysis of the sensitivity of the performance with the choice of the slack parameter $\delta$ and initialization was performed under the following setting: models were trained for 50 epochs on MNIST with hypervolume maximization using 8, 16, 24 discriminators. Three independent runs (different initializations) were executed with each $\delta = \{1.05, 1.5, 1.75, 2\}$ and number of discriminators, totalizing 36 final models. Fig. 5 reports the box-plots obtained for 5 FID independent computations using 10000 images, for each of the 36 models obtained under the setting previously described. Results clearly indicate that increasing the number of discriminators yields much smaller variation in the FID obtained by the final model.

---

[2]https://docs.scipy.org/doc/scipy/reference/tutorial/optimize.html

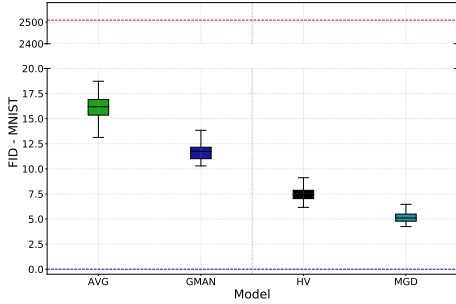

Figure 2: Box-plots corresponding to 30 independent FID computations with 10000 images. MGD performs consistently better than other methods, followed by hypervolume maximization. Models that achieved minimum FID at train time were used. Red and blue dashed lines are the FIDs of a random generator and real data, respectively.

Figure 3: Time vs. best FID achieved during training for each approach. FID values are computed over 1000 generated images after every epoch. MGD performs relevantly better than others in terms of FID, followed by HV. However, MGD is approximately 7 times slower than HV. HV is well-placed in the time-quality trade-off.

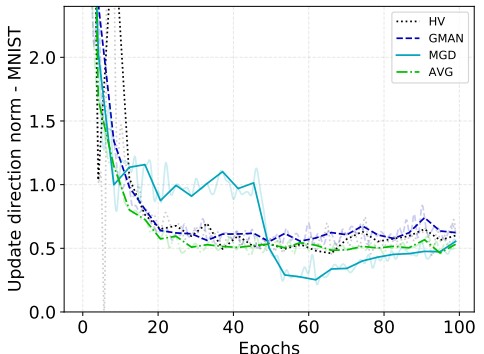

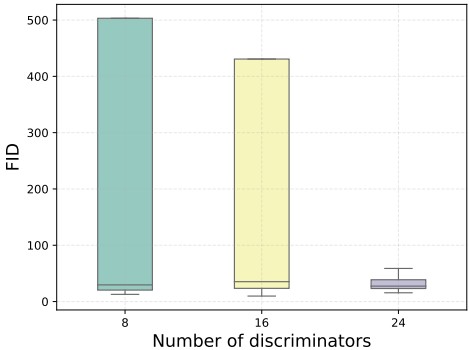

Figure 4: Norm of the update direction over time for each method. While Pareto-stationarity is approximately achieved by all methods, performance varies relevantly in terms of FID.

Figure 5: Independent FID evaluations for models obtained with different runs using distinct slack parameter $\delta$. Sensitivity reduces as the number of discriminators increases.

## 5.2 HV AS AN ALTERNATIVE FOR MGD

We evaluate the performance of HV compared to baseline methods using the CIFAR-10 dataset. FID was computed with a pretrained ResNet (He et al., 2016). ResNet was trained on the 10-class classification task of CIFAR-10 up to approximately $95\%$ test accuracy. DCGAN (Radford et al., 2015) and WGAN-GP (Gulrajani et al., 2017) were included in the experiments for FID reference. Same architectures as in (Neyshabur et al., 2017) were employed for all multi-discriminators settings. An increasing number of discriminators was used. Inception score as well as FID computed with other models are included in Appendix C.

In Fig. 6, we report the box-plots of 15 independent evaluations of FID on 10000 images for the best model obtained with each method across 3 independent runs. Results once more indicate that HV outperforms other methods in terms of quality of the generated samples. Moreover, performance clearly improves as the number of discriminators grows. Fig. 7 shows the FID at train time, i.e. measured with 1000 generated samples after each epoch, for the best models across runs. Models trained against more discriminators clearly converge to smaller values. We report the norm of the update direction $|| \sum_{k=1}^{K} \alpha_k \nabla l_k ||$ for each method in Fig. 9, Appendix C.

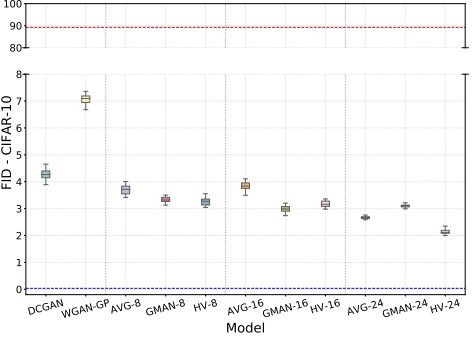 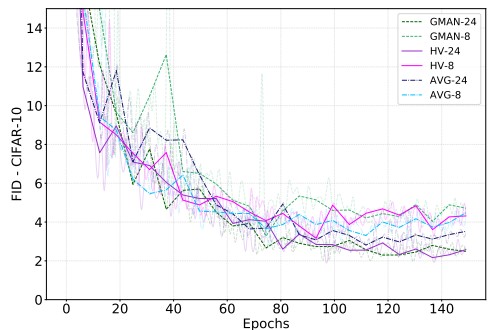

Figure 6: Box-plots of 15 independent FID computations with 10000 images. Dashed lines are real data (blue) and random generator (red) FIDs.

Figure 7: FID estimated over 1000 generated images at train time. Models trained against more discriminators achieve lower FID.

**Cost under the multiple discriminator setting.** We highlight that even though training with multiple discriminators may be more computationally expensive when compared to conventional approaches, such framework supports fully parallel training of the discriminators, a feature which is not trivially possible in other GAN settings. For example in WGAN, the discriminator is serially updated multiple times for each generator update. In Fig. 10 at Appendix C, we provide a comparison between the wall-clock time per iteration between all methods evaluated. Serial implementations of discriminators updates with 8 and 16 discriminators were faster than WGAN-GP.

## 5.3 Effect of the number of discriminators on sample diversity

We repeat the experiments in (Srivastava et al., 2017) aiming to analyze how the number of discriminators impacts the sample diversity of the corresponding generator when trained using hypervolume maximization. The stacked MNIST dataset is employed and results reported in (Lin et al., 2017) are used for comparison. HV results for 8, 16, and 24 discriminators were obtained with 10k and 26k generator images averaged over 10 runs. The number of covered modes along with the KL divergence between the generated mode distribution and test data are reported in Table 1.

| Test samples | Model | Modes (Max 1000) | KL |
|---|---|---|---|
| | DCGAN (Radford et al., 2015) | 99.0 | 3.400 |
| | ALI (Dumoulin et al., 2016) | 16.0 | 5.400 |
| 26k | Unrolled GAN (Metz et al., 2016) | 48.7 | 4.320 |
| | VEEGAN (Srivastava et al., 2017) | 150.0 | 2.950 |
| | PacDCGAN2 (Lin et al., 2017) | $1000.0 \pm 0.0$ | $0.060 \pm 0.003$ |
| | HV - 8 disc. | $679.2 \pm 5.9$ | $1.139 \pm 0.011$ |
| 10k | HV - 16 disc. | $998.0 \pm 1.8$ | $0.120 \pm 0.004$ |
| | HV - 24 disc. | $998.3 \pm 1.1$ | $0.116 \pm 0.003$ |
| | HV - 8 disc. | $776.8 \pm 6.4$ | $1.115 \pm 0.007$ |
| 26k | HV - 16 disc. | $1000.0 \pm 0.0$ | $0.088 \pm 0.002$ |
| | HV - 24 disc. | $1000.0 \pm 0.0$ | $0.084 \pm 0.002$ |

Table 1: Number of covered modes and reverse KL divergence for stacked MNIST.

As in previous experiments, results improved as we increased the number of discriminators. All evaluated models using HV outperformed DCGAN, ALI, Unrolled GAN and VEEGAN. Moreover, HV with 16 and 24 discriminators achieved state-of-the-art coverage values. Thus, the increase in models' capacity via using more discriminators directly resulted in an improvement in generator's coverage. Training details as well as architectures information are presented in Appendix B.

## 6 Conclusion

In this work we have shown that employing multiple discriminators is a practical approach allowing us to trade extra capacity, and thereby extra computational cost, for higher quality and diversity of generated samples. Such an approach is complimentary to other advances in GANs training and can be easily used together with other methods. We introduced a multi-objective optimization framework for studying multiple discriminator GANs, and showed strong similarities between previous work and the multiple gradient descent algorithm. The proposed approach was observed to consistently yield higher quality samples in terms of FID. Furthermore, increasing the number of discriminators was shown to increase sample diversity and generator robustness.

Deeper analysis of the quantity $|| \sum_{k=1}^{K} \alpha_k \nabla l_k ||$ is the subject of future investigation. We hypothesize that using it as a penalty term might reduce the necessity of a high number of discriminators.

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

# APPENDIX

## A - OBJECTIVE EVALUATION METRIC.

In (Heusel et al., 2017), authors proposed to use as a quality metric the squared Fréchet distance (Fréchet, 1957) between Gaussians defined by estimates of the first and second order moments of the outputs obtained through a forward pass in a pretrained classifier of both real and generated data. They proposed the use of Inception V3 (Szegedy et al., 2016) for computation of the data representation and called the metric Fréchet Inception Distance (FID), which is defined as:

$$\text{FID} = ||m_d - m_g||^2 + \text{Tr}(\Sigma_d + \Sigma_g - 2(\Sigma_d \Sigma_g)^{\frac{1}{2}}), \tag{10}$$

where $m_d, \Sigma_d$ and $m_g, \Sigma_g$ are estimates of the first and second order moments from the representations of real data distributions and generated data, respectively.

We employ FID throughout our experiments for comparison of different approaches. However, for each dataset in which FID was computed, the output layer of a pretrained classifier on that particular dataset was used instead of Inception. $m_d$ and $\Sigma_d$ were estimated on the complete test partitions, which are not used during training.

## B - EXPERIMENTAL SETUP FOR STACKED MNIST EXPERIMENTS AND GENERATOR'S SAMPLES

Architectures of the generator and discriminator are detailed in Tables 2 and 3, respectively. Batch normalization was used in all intermediate convolutional and fully connected layers of both models. We employed RMSprop to train all the models with learning rate and $\alpha$ set to $0.0001$ and $0.9$, respectively. Mini-batch size was set to $64$. The setup in (Lin et al., 2017) is employed and we build 128000 and 26000 samples for train and test sets, respectively.

| Layer | Outputs | Kernel size | Stride | Activation |
|-------|---------|-------------|--------|------------|
| Input: $z \sim \mathcal{N}(0, I_{100})$ | | | | |
| Fully connected | 2*2*512 | 4, 4 | 2, 2 | ReLU |
| Transposed convolution | 4*4*256 | 4, 4 | 2, 2 | ReLU |
| Transposed convolution | 8*8*128 | 4, 4 | 2, 2 | ReLU |
| Transposed convolution | 14*14*64 | 4, 4 | 2, 2 | ReLU |
| Transposed convolution | 28*28*3 | 4, 4 | 2, 2 | Tanh |

Table 2: Generator's architecture.

| Layer | Outputs | Kernel size | Stride | Activation |
|-------|---------|-------------|--------|------------|
| Input | 28*28*3 | | | |
| Projection | 14*14*3 | 8, 8 | 2, 2 | |
| Convolution | 7*7*64 | 4, 4 | 2, 2 | LeakyReLU |
| Convolution | 5*5*128 | 4, 4 | 2, 2 | LeakyReLU |
| Convolution | 2*2*256 | 4, 4 | 2, 2 | LeakyReLU |
| Convolution | 1 | 4, 4 | 2, 2 | Sigmoid |

Table 3: Discriminator's architecture.

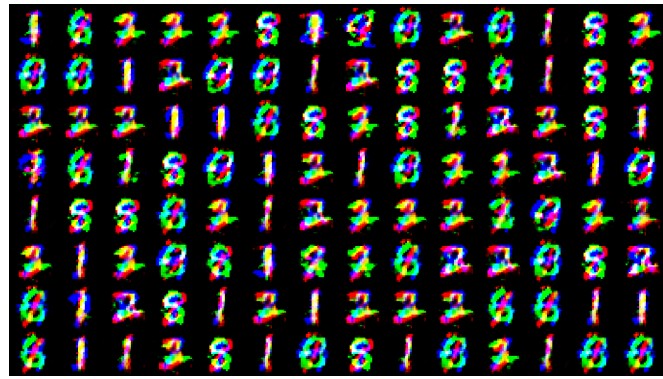

(a) HV - 8 discriminators

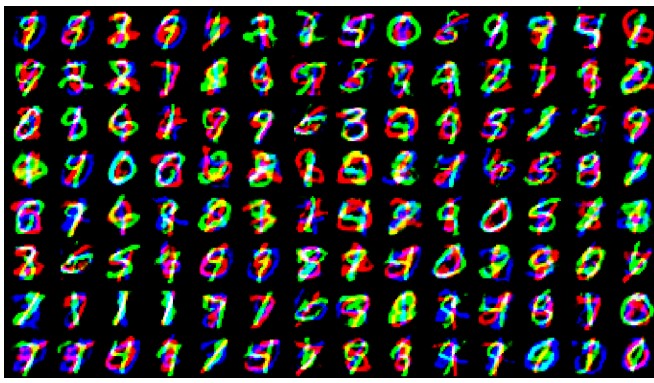

(b) HV - 16 discriminators

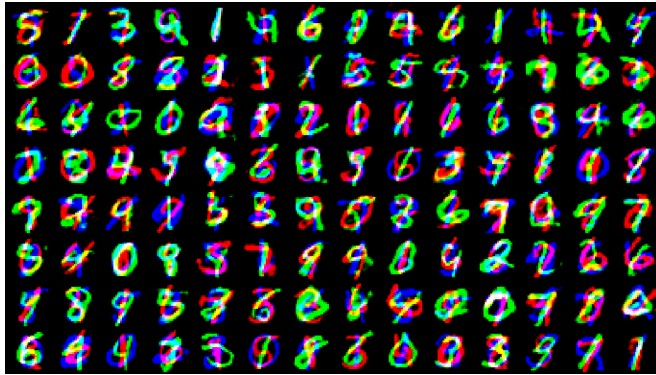

(c) HV - 24 discriminators

Figure 8: Stacked MNIST samples for HV trained with 8, 16, and 24 discriminators. Samples diversity increases greatly when more discriminators are employed.

## C - EXTRA RESULTS ON CIFAR-10

### C.1 - MULTIPLE DISCRIMINATORS ACROSS DIFFERENT INITIALIZATIONS AND OTHER SCORES

Table 4 presents the best FID (computed with a pretrained ResNet) achieved by each approach at train time, along with the epoch in which it was achieved, for each of 3 independent runs. Train time FIDs are computed using 1000 generated images.

| #D | Method | Best FID (epoch) |
|---|---|---|
| 1 | DCGAN | 7.09 (68), 9.09 (21), 4.22 (101) |
|  | WGAN-GP | 5.09 (117), 5.69 (101) 7.13 (71) |
| 8 | AVG | 3.35 (105), 4.64 (141), 3.00 (76) |
|  | GMAN | 4.28 (123), 4.24 (129), 3.80 (133) |
|  | HV | 3.87 (102), 4.54 (82), 3.20 (98) |
| 16 | AVG | 3.16 (96), 2.50 (91), 2.77 (116) |
|  | GMAN | 2.69 (129), 2.36 (144), 2.48 (120) |
|  | HV | 2.56 (85), 2.70 (97), 2.68 (133) |
| 24 | AVG | 2.10 (94), 2.44 (132), 2.43 (129) |
|  | GMAN | 2.16 (120), 2.02 (98), 2.13 (130) |
|  | HV | 2.05 (83), 1.89 (97), 2.23 (130) |

Table 4: Best FID obtained for each approach on 3 independent runs. FID is computed on 1000 generated images after every epoch.

In Fig. 9, we report the norm of the update direction $||\sum_{k=1}^{K} \alpha_k \nabla l_k||$ of the best model obtained for each method. Interestingly, different methods present similar behavior in terms of convergence in the Pareto-stationarity sense, i.e. the norm upon convergence is lower for models trained against more discriminators, regardless of the employed method.

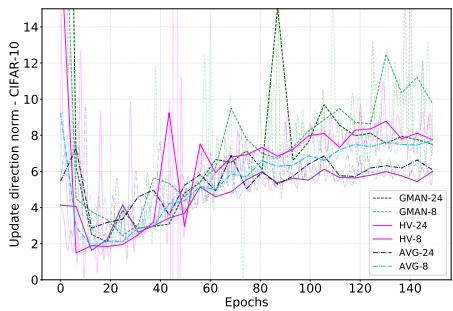

Figure 9: Norm of the update direction over time for each method. Higher number of discriminators yield lower norm upon convergence.

We computed extra scores using 10000 images generated by the best model reported in Table 4, i.e. the same models utilized to generate the results shown in Fig. 6. Both Inception score and FID were computed with original implementations, while FID-VGG and FID-ResNet were computed using a VGG and a ResNet we pretrained. Results are reported with respect to DCGAN's scores.

|  | WGAN-GP | AVG-8 | AVG-16 | AVG-24 | GMAN-8 | GMAN-16 | GMAN-24 | HV-8 | HV-16 | HV-24 |
|---|---|---|---|---|---|---|---|---|---|---|
| *Inception Score* | 1.08 | 1.02 | 1.26 | 1.36 | 0.95 | 1.32 | 1.42 | 1.00 | 1.30 | **1.44** |
| *FID* | 0.80 | 0.98 | 0.76 | 0.73 | 0.92 | 0.79 | **0.65** | 0.89 | 0.77 | 0.72 |
| *FID-VGG* | 1.29 | 0.91 | 1.03 | 0.85 | 0.87 | 0.78 | 0.73 | 0.78 | 0.75 | **0.64** |
| *FID-ResNet* | 1.64 | 0.88 | 0.90 | 0.62 | 0.80 | 0.72 | 0.73 | 0.75 | 0.73 | **0.51** |

Table 5: Scores of different methods measure on generated CIFAR-10 samples. DCGAN scores are used as reference values, and results report are the ratio between given model and DCGAN scores. Inception score is better when high, whereas FIDs are better when low.

### C.2 - COMPUTATIONAL COST

In Table 6 we present a comparison of minimum FID-ResNet obtained during training, along with computation cost in terms of time and space for different GANs, with both 1 and 24 discriminators. The computational cost of training GANs under a multiple discriminator setting is higher by design, in terms of both FLOPS and memory, if compared with single discriminators settings. However, a corresponding shift in performance is the result of the additional cost. This effect was consistently observed considering 4 different well-known approaches, namely DCGAN (Radford et al., 2015), Least-square GAN (LSGAN) (Mao et al., 2017), and HingeGAN (Miyato et al., 2018). The architectures of all single discriminator models follow the DCGAN, described in (Radford et al., 2015). For the 24 discriminators models, we used the architecture described in (Neyshabur et al., 2017), which consists in removing the the normalization layers from DCGAN's discriminator and further adding the projection layer, inline with previous experiments reported for CIFAR-10 upscaled to 64x64. All models were trained with minibatch size of 64 during 150 epochs. Adam (Kingma & Ba, 2014) was used as the optimizer. Learning rate, $\beta_1$ and $\beta_2$ were equal to 0.0002, 0.5 and 0.999, respectively.

|  | # Discriminators | FID-ResNet | FLOPS (MAC) | Memory (Mb) |
|---|---|---|---|---|
| DCGAN | 1 | 4.22 | 8e10 | 1292 |
|  | 24 | 1.89 | 5e11 | 5671 |
| LSGAN | 1 | 4.55 | 8e10 | 1303 |
|  | 24 | 1.91 | 5e11 | 5682 |
| HingeGAN | 1 | 6.17 | 8e10 | 1303 |
|  | 24 | 2.25 | 5e11 | 5682 |

Table 6: Comparison between different GANs with 1 and 24 discriminators in terms of minimum FID-ResNet obtained during training, and FLOPs and memory consumption for a complete train step.

Furthermore, wall-clock time per iteration for different numbers of discriminators is shown in Fig. 10 for experiments with CIFAR-10 with serial updates of discriminators. Notice that while the increase in cost in terms of FLOPS and memory is unavoidable when multiple discriminators settings is employed, wall-clock time can be made close to single discriminators cases since training with respect to different discriminators can be implemented in parallel. On the other hand, extra cost in time introduced by other frameworks such as WGAN-GP or SNGAN cannot be trivially recovered.

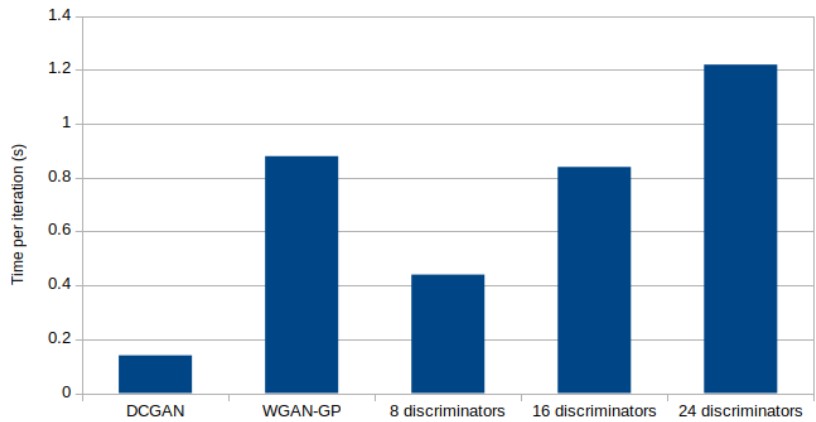

Figure 10: Time in seconds per iteration of each method for serial updates of discriminators. Multiple discriminators approaches considered do not present relevant difference in time per iteration.

## C.3 - GENERATED SAMPLES

In Figs. 11, 12, and 13 we show random generated samples with 8, 16, and 24 discriminators for AVG, GMAN, and HV, respectively.

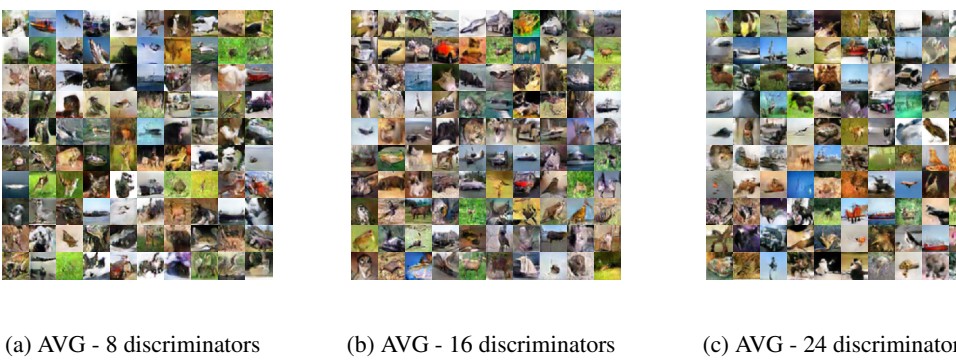

(a) AVG - 8 discriminators     (b) AVG - 16 discriminators     (c) AVG - 24 discriminators

Figure 11: CIFAR-10 samples for AVG trained with 8, 16, and 24 discriminators.

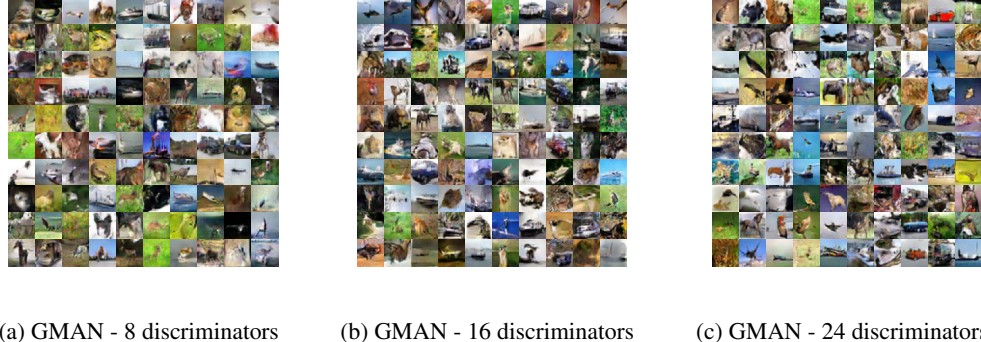

(a) GMAN - 8 discriminators     (b) GMAN - 16 discriminators     (c) GMAN - 24 discriminators

Figure 12: CIFAR-10 samples for GMAN trained with 8, 16, and 24 discriminators.

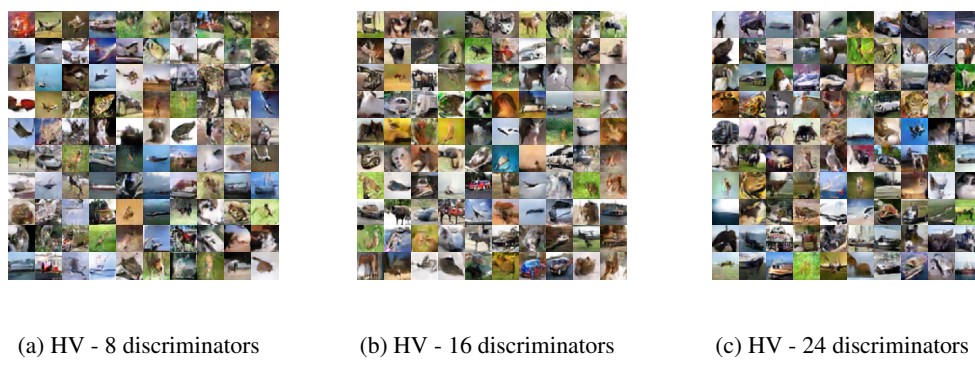

(a) HV - 8 discriminators     (b) HV - 16 discriminators     (c) HV - 24 discriminators

Figure 13: CIFAR-10 samples for HV trained with 8, 16, and 24 discriminators.

C.4 - RESULTS CIFAR-10 32x32

All results reported in previous sections using CIFAR-10 were obtained with an upscaled version of the dataset. Here, we thus run experiments with the dataset in its original resolution aiming to contextualize our proposed approach with respect to previously introduced methods. To do so, we repeated similar experiments as reported in Miyato et al. (2018)-Table 2, for the model referred to as *standard CNN*. The same architecture is employed and the spectral normalization is removed from the discriminators. Moreover, the same projection input is added in each of the discriminators.

Results in terms of both FID and Inception score, evaluated on top of 5000 generated images as in (Miyato et al., 2018) as well as with 10000 images, are reported in Table 7 for our proposed approach and our implementation of (Miyato et al., 2018), along with the FID measured using a ResNet classifier trained in advance.

As can be seen, the addition of the multiple discriminators setting along with hypervolume maximization yields a relevant shift in performance for the DCGAN-like generator, taking all evaluated metrics to levels of recently proposed GANs.

|  | FID-ResNet | FID (5k) | IS (5k) | FID (10k) | IS (10k) |
|---|---|---|---|---|---|
| SNGAN (Miyato et al., 2018) | - | 25.5 | $7.58 \pm 0.12$ | - | - |
| WGAN-GP (Miyato et al., 2018) | - | 40.2 | $6.68 \pm 0.06$ | - | - |
| DCGAN (Miyato et al., 2018) | - | - | $6.64 \pm 0.14$ | - | - |
| SNGAN (our implementation) | 1.55 | 27.93 | $7.11 \pm 0.30$ | 25.29 | $7.26 \pm 0.12$ |
| DCGAN + 24 Ds and HV | 1.21 | 27.74 | $7.32 \pm 0.26$ | 24.90 | $7.45 \pm 0.17$ |

Table 7: Evaluation of the effect of adding discriminators on a DCGAN-like model trained on CIFAR-10. Results reach the same level as the best reported for the given architecture when the multiple-discriminator setting is added and the normalization layers are removed from discriminators.

## D - CELEBA DATASET

### D.1 - COMPARING WITH OTHER MULTIPLE-DISCRIMINATORS APPROACHES

Here, we present samples obtained by generators trained against 8, 16, and 24 discriminators using AVG, GMAN, and HV on the CelebA dataset rescaled to 64x64. Training lasted 100 epochs and samples are shown in Figs. 14, 15, and 16 for AVG, GMAN and HV, respectively. Same architectures and hyperparameters used for experiments with CIFAR-10 presented in Section 5 were utilized.

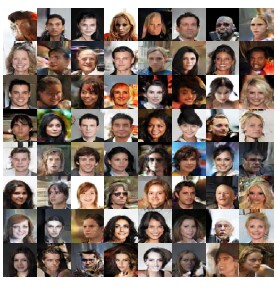
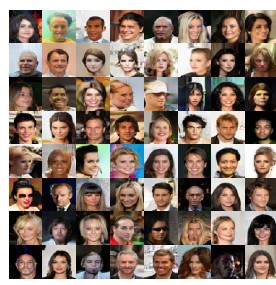

(a) AVG - 8 discriminators          (b) AVG - 16 discriminators          (c) AVG - 24 discriminators

Figure 14: CelebA samples for AVG trained with 8, 16, and 24 discriminators.

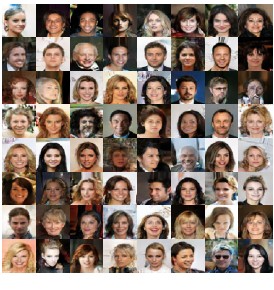
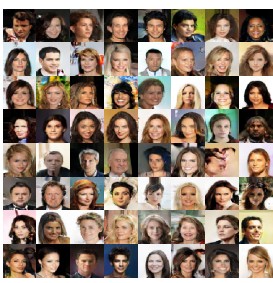
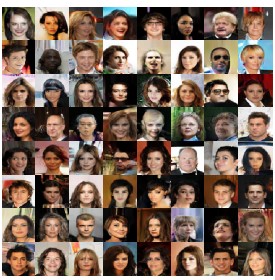

(a) GMAN - 8 discriminators        (b) GMAN - 16 discriminators        (c) GMAN - 24 discriminators

Figure 15: CelebA samples for GMAN trained with 8, 16, and 24 discriminators.

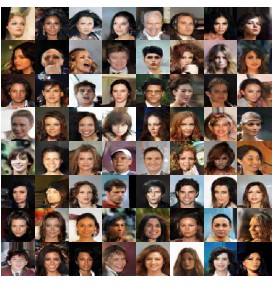
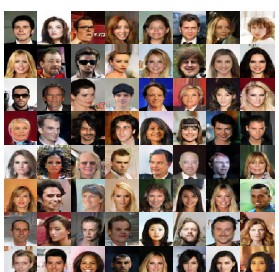

(a) HV - 8 discriminators            (b) HV - 16 discriminators            (c) HV - 24 discriminators

Figure 16: CelebA samples for HV trained with 8, 16, and 24 discriminators.

### D.2 - GENERATING 128x128 IMAGES

In this experiment, we verify whether the proposed multiple discriminators setting is capable of generating higher resolution images. For that, we employed the CelebA at a size of 128x128. We used a similar architecture for both generator and discriminators networks as described in the previous experiments. A convolutional layer with 2048 feature maps was added to both generator and discriminators architectures due to the increase in the image size. Adam optimizer with the same set of hyperparameters as for CIFAR-10 and CelebA 64x64 was employed. We trained models with 6, 8, and 10 discriminators during 24 epochs. Samples from each generator are shown in Figure 17.

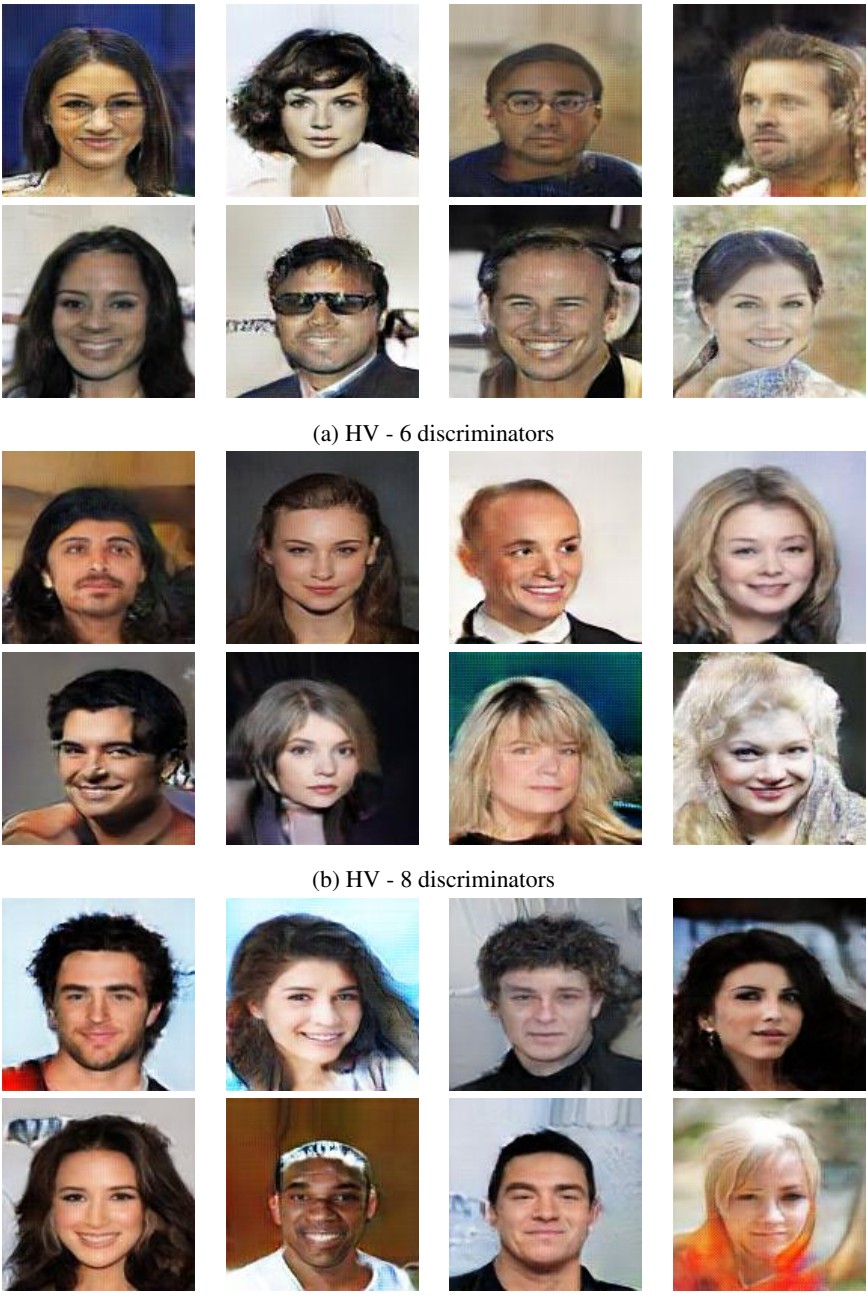

(a) HV - 6 discriminators

(b) HV - 8 discriminators

(c) HV - 10 discriminators

Figure 17: 128x128 CelebA samples for HV trained during 24 epochs with 6, 8, and 10 discriminators.

## E - GENERATING 256X256 CATS

We show the proposed multiple-discriminators setting scales to higher resolution even in the small dataset regime, by reproducing the experiments presented in (Jolicoeur-Martineau, 2018). We used the same architecture for the generator. For the discriminator, we removed batch normalization from all layers and used stride equal to 1 at the last convolutional layer, after adding the initial projection step. The Cats dataset [3] was employed, we followed the same pre-processing steps, which, in our case, yielded 1740 training samples with resolution of 256x256. Our model is trained using 24 discriminators and Adam optimizer with the same hyperparameters as for CIFAR-10 and CelebA previously described experiments. In Figure 18 we show generator's samples after 288 training epochs. One epoch corresponds to updating over 27 minibatches of size 64.

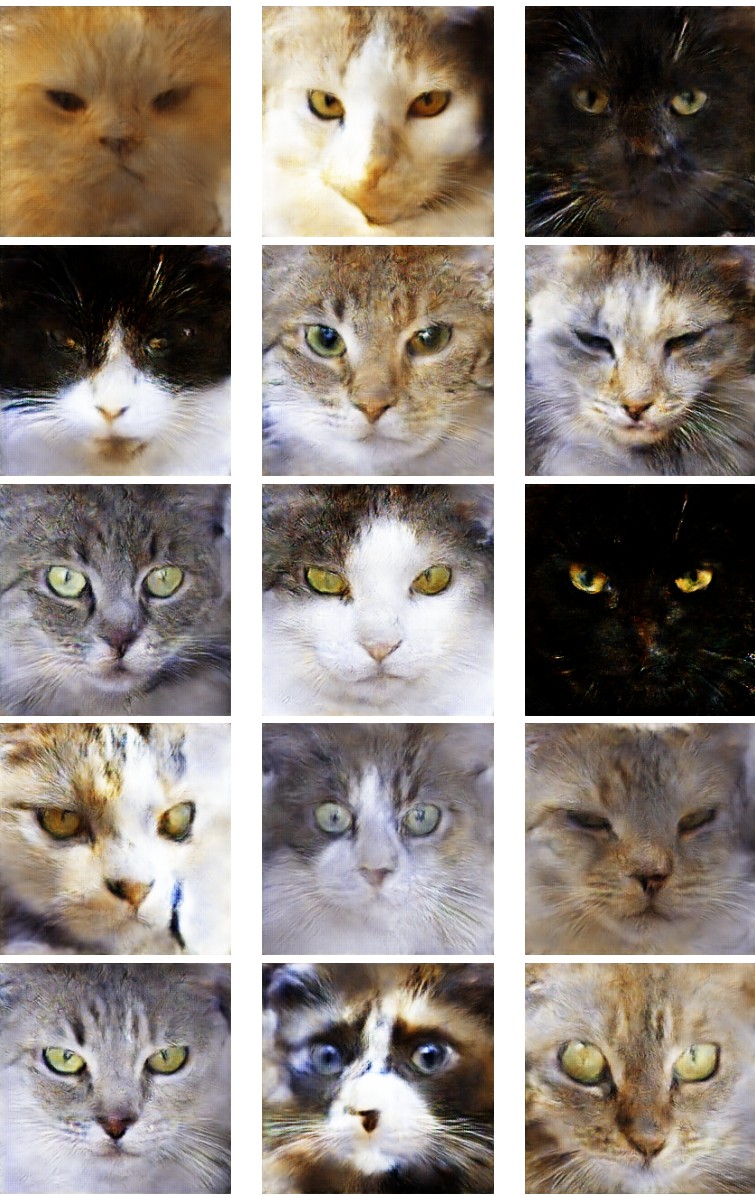

Figure 18: Cats generated using 24 discriminators after 288 training epochs.

---

[3]https://www.kaggle.com/crawford/cat-dataset

## F - INCREASING NUMBER OF RANDOM PROJECTIONS

In this experiment we illustrate and confirm the results introduced in (Neyshabur et al., 2017), showing the effect of using an increasing number of random projections to train a GAN. We trained models using average loss minimization with 1 to 6 discriminators on the CelebA dataset for 15 epochs. Samples from the generator obtained in the last epoch are shown in Fig. 19. Generated samples are closer to real data as the number of random projections (and discriminators, consequently) increases.

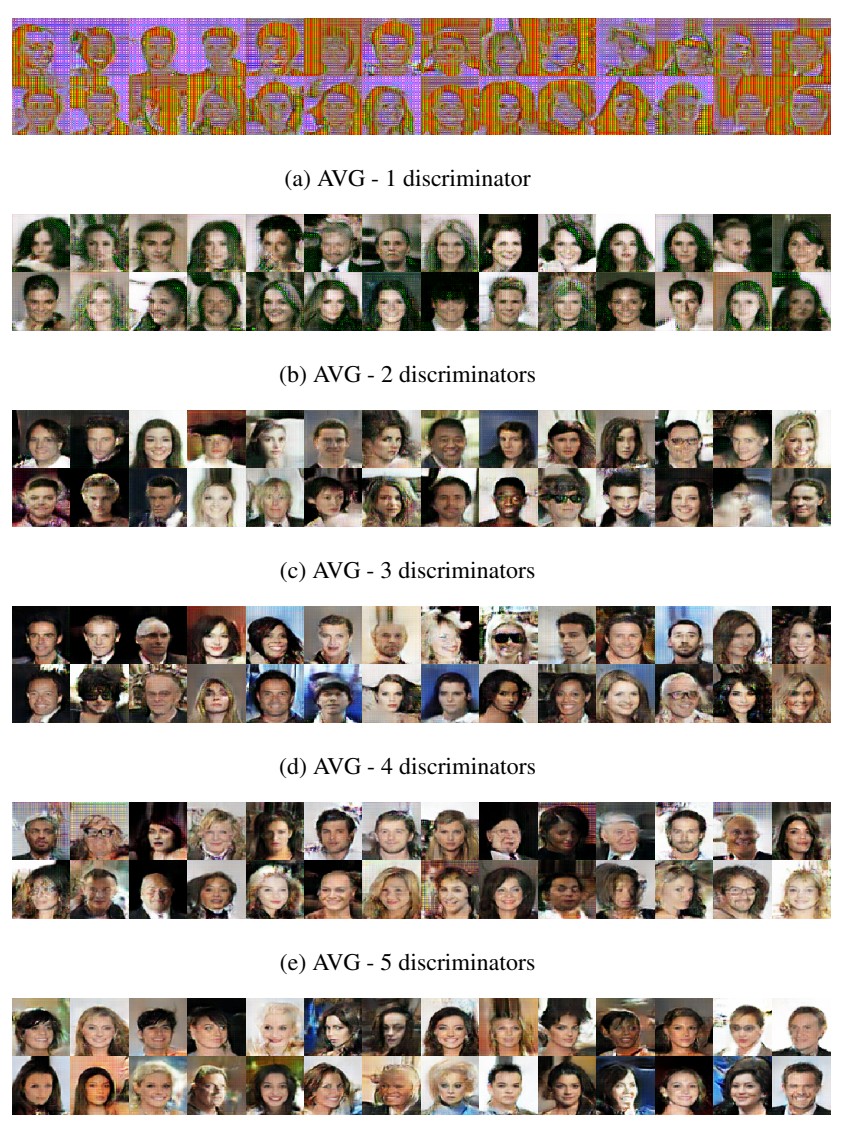

(a) AVG - 1 discriminator

(b) AVG - 2 discriminators

(c) AVG - 3 discriminators

(d) AVG - 4 discriminators

(e) AVG - 5 discriminators

(f) AVG - 6 discriminators

Figure 19: Models trained with AVG during 15 epochs using an increasing number of random projections and discriminators.

# G - Illustration of Interaction Between Hypervolume and Adopted Nadir Point Adaptation Scheme

Consider a two-objectives problem, with $l_1^t > 0$ and $l_2^t > 0$ corresponding to each of the losses we want to minimize, at iteration $t$. We present in Figures 20 and 21 an illustrative example of the effect of the adaptation scheme adopted for $\eta$, as described in Section 4.

Figure 20 describes the initialization state. Since $l_1^t$ and $l_2^t$ will be high at $t = 0$, and, following the adaptation rule presented in previous sections, $\eta^t = \delta \max\{l_1^t, l_2^t\}$, for a *slack* $\delta > 0$, the difference $\eta^t - \max\{l_1^t, l_2^t\}$ will be high. In contrast, after $T$ updates, as described in Figure 21, $\eta^t = \delta \max\{l_1^t, l_2^t\}$ will be smaller, since losses are now closer to 0.

If no adaptation is performed and $\eta$ is kept unchanged throughout training, as represented in red in Figure 21, $\eta^T - l_1^T \approx \eta^T - l_2^T$ for a large enough $T$, which will end up assigning similar weights to gradients provided by the different losses, defeating the purpose of employing hypervolume maximization rather than optimizing for the average loss.

The employed adaptation scheme thus keeps the gradient weighting relevant even when losses become low. Moreover, this effect will be more aggressive as training progresses, assigning more gradient importance to the higher losses, since $\eta^T - \max\{l_1^T, l_2^T\} < \eta^0 - \max\{l_1^0, l_2^0\}$.

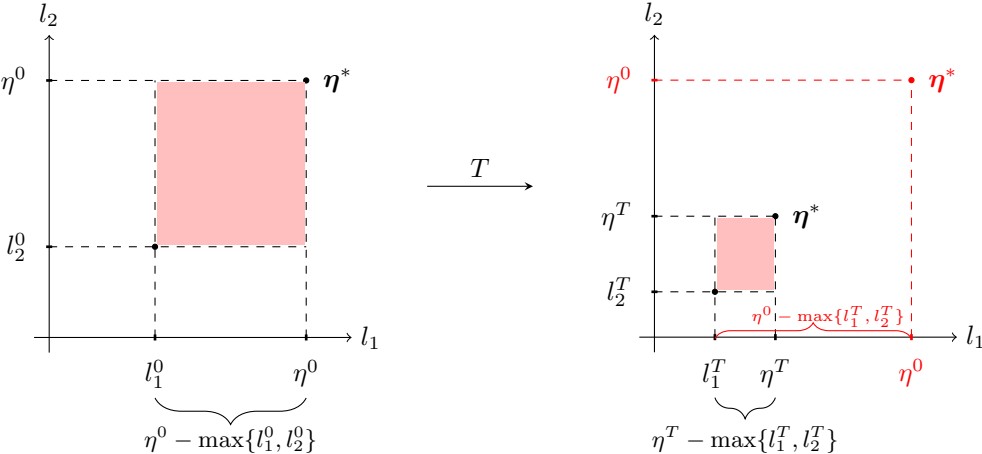

Figure 20: Losses and nadir point at beginning of training.

Figure 21: Losses and nadir point at $t = T$, and nadir point at $t = 0$ (in red).

## H - WALL-CLOCK TIME FOR REACHING BEST FID DURING TRAINING ON MNIST

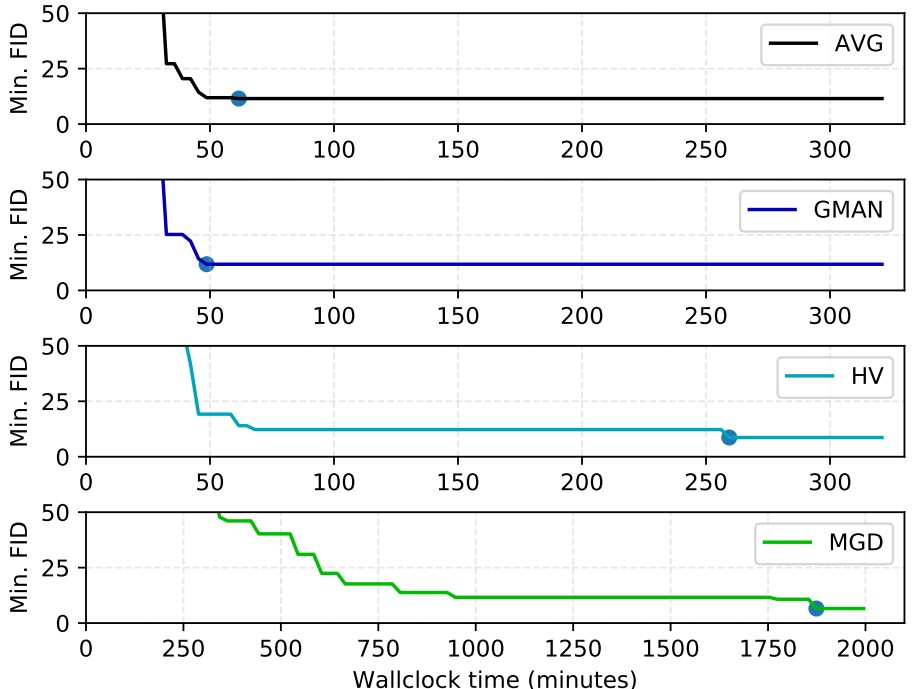

Figure 22: Minimum FID during training. X-axis is in minutes. The blue dot is intended to highlight the moment during training when the minimum FID was reached.

