# OpenReview forum: "Multi-objective training of Generative Adversarial Networks with multiple discriminators"
_ICLR.cc/2019/Conference_

### Official Review · AnonReviewer2 · 2018-11-02
**interesting methods, ok results**

**Rating:** 6
**Confidence:** 4

**Review:**

The paper investigates the use of multi-objective optimization techniques in GAN-setups where there are multiple discriminators. Using multiple discriminators was proposed in Durugkar et al, Arora et al, Neyshabur et al and others. The twist here is to focus on the Pareto front and to import multiple gradient descent and hypervolume-maximization based methods into GANs.

The results are decent. The authors find that optimizing with respect to multiple discriminators increases diversity of samples for a computational cost. However, just scaling up (and carefully optimizing), can yield extremely impressive samples, https://arxiv.org/abs/1809.11096. It is unclear how the tradeoffs in optimizing against multiple discriminators stack-up against bigger GANs.

From my perspective, the paper is interesting because it introduces new methods into GANs from another community. However, the results themselves are not sufficient for publication.

---

> ### Author Response · Authors · 2018-11-10
> **Response to Reviewer 2**
>
> We thank the reviewer for the feedback and taking the time for reading our paper. We are glad that the reviewer found our method interesting and hope that the following response, added to the new results included in the manuscript, will make her/him more confident about our contributions.
>
> Regarding the mentioned trade-off, we understood that the reviewer is referring to the addition of “capacity” (in terms of the number of parameters) on the discriminators side, as in the multiple-discriminators settings, or in the generator side, as in the pointed reference (Brock et al. 2018). We would like to point out that such approaches have different goals: while adding capacity in the generator side is intended to yield generators able to scale to higher resolution settings and higher quality samples, multiple-discriminators are aimed at stabilizing training, avoiding common issues such as mode-collapse and divergence, which makes final performance of the generator highly dependent of careful hyperparameters tuning. If enough resources are available, both approaches should be used jointly. As we also observed multiple-discriminators training to yield higher quality and diversity when compared to their single-discriminator equivalents, we believe higher scales settings would also benefit.
>
> Regarding the insufficiency of results, we would like to respectfully highlight that we presented quantitative and qualitative results (comparing with both single- and multiple-discriminators GANs) in 4 datasets (namely, MINIST, CIFAR-10, Stacked MNIST, and CelebA), with consistent conclusions. We also included samples on a higher resolution for CelebA at 128x128 in Appendix D.2, and are currently running experiments to compare different versions of GANs in the single  vs. multiple discriminators settings. Some preliminary results which will be added to the manuscript as soon as we conclude the new experiments, show that adding discriminators yield the following relative improvement in terms of FID: DCGAN - 55.21%; LSGAN - 57.93% (we are currently running similar experiments on other GANs such as wGAN-GP and hingeGAN [2]). Moreover, we would highly appreciate if the reviewer could suggest any further experiment in order to increase her/his confidence in our results.
>
> [2] Miyato, Takeru, et al. "Spectral normalization for generative adversarial networks." arXiv preprint arXiv:1802.05957 (2018).

---

> > ### Comment · AnonReviewer2 · 2018-11-17
> > **comment**
> >
> > 1. I didn't articulate my comment about tradeoffs well; it's been partially addressed by App C.2
> >
> > 2. The idea of using multi-objective optimization to improve stability is really interesting. Maybe I missed something, but there seems to be a large jump from "GANs are unstable" to "multi-objective optimization should help". Is there anything (say, a theoretical result or conceptual explanation) in the literature to fill the gap?

---

> > > ### Author Response · Authors · 2018-11-17
> > > **Response to comment**
> > >
> > > Thank you for your comment and for your time reading the updated version of our manuscript.
> > >
> > > “2. The idea of using multi-objective optimization to improve stability is really interesting. Maybe I missed something, but there seems to be a large jump from "GANs are unstable" to "multi-objective optimization should help". Is there anything (say, a theoretical result or conceptual explanation) in the literature to fill the gap?”
> > >
> > > **For the “GANs are unstable” part, we build upon two results introduced in [1], namely:
> > >
> > > 1-In Theorem A.1, it is shown that marginals along random projections will likely have a higher overlap. This means that, in the projected space, generator’s and real samples will be more alike. This avoids a common failure mode in GANs training which corresponds to when the discriminators quickly learn how to distinguish real and generated samples. Thus, training in a lower-dimensional randomly projected space will be easier if compared to the original data.
> > >
> > > 2- In Theorem A.2, an upper-bound is proven to show that if approximation of the projected data distribution is achieved along a sufficient number of random projections (each corresponding to a discriminator), the distribution induced by the generator approximates the real data distribution (in the original space).
> > >
> > > As we see, authors in [1] propose to trade one hard problem for a number of easier subproblems in order to ameliorate training instability.
> > >
> > > **Regarding the "multi-objective optimization should help" aspect, we proposed a more suitable optimization framework to be used in the described setting looking at it through the lens of multi-objective optimization, by applying the hypervolume maximization approach. We compared previously proposed multiple-discriminators approaches as well as Multiple Gradient Descent, and showed hypervolume maximization to yield a better compromise between cost in time and sample quality.
> > >
> > > We further respectfully point the reviewer to the last paragraph of Section 4.1, which we post herein for convenience. There, we aim at further motivate our contribution by comparing the proposed method with simply minimizing for the average loss:
> > >
> > > “The upper bound proven by [1] assumes that the marginals of the real and generated distributions are identical along all random projections. Average loss minimization does not ensure equally good approximation between the marginals along all directions. In case of a trade-off between discriminators, i.e. if decreasing the loss on a given projection increases the loss with respect to another one, the distribution of losses can be uneven. With HV on the other hand, especially when \eta is reduced throughout training, overall loss will be kept high as long as there are discriminators with high loss. This objective tends to prefer central regions of a trade-off, in which all discriminators present a roughly equally low loss.”
> > >
> > > We hope to have appropriately clarified your concerns.
> > >
> > > [1] Neyshabur, Behnam, Srinadh Bhojanapalli, and Ayan Chakrabarti. "Stabilizing GAN training with multiple random projections." arXiv preprint arXiv:1705.07831 (2017).

---

### Official Review · AnonReviewer3 · 2018-11-03
**The idea is natural and interesting, the presentation is clear, but short of analysis on the computational cost (FLOPS and memory consumption)**

**Rating:** 5
**Confidence:** 3

**Review:**

This paper studies the problem of training of Generative Adversarial Networks employing a set of discriminators, as opposed to the traditional game involving one generator against a single model. Specifically, this paper claims two contributions:
1.	We offer a new perspective on multiple-discriminator GAN training by framing it in the context of multi-objective optimization, and draw similarities between previous research in GANs variations and MGD, commonly employed as a general solver for multi-objective optimization.
2.	We propose a new method for training multiple-discriminator GANs: Hypervolume maximization, which weighs the gradient contributions of each discriminator by its loss.

Overall, the proposed method is empirical and the authors show its performance by experiments.

First, I want to discuss the significance of this work (or this kind of work). As surveyed in the paper, the idea of training of Generative Adversarial Networks employing a set of discriminators has been explored by several previous work, and showed some performance improvement. However, this idea (methods along this line) is not popular in GAN applications, like image-to-image translation. I guess that the reason may be that: the significant computational cost (both in FLOPS and memory consumption) increase due to multiple discriminators destroys the benefit from the small performance improvement. Maybe I’m wrong. In Appendix C Figure 10, the authors compares the wall-lock time between DCGAN, WGAN-GP and multiple-discriminator, and claims that the proposed approach is cheaper than WGAN-GP. However, WGAN-GP is more expensive due to its loss function involves gradients, while the proposed method does not. If directly compared with DCGAN, we can see an obvious increase in wall-clock time (FLOPS). In addition, the additional memory consumption is hidden there, which is a bigger problem in practice when the discriminators are large. SN-GAN have roughly the same computational cost and memory consumption of DC-GAN, but inception and FID are much higher. From my perspective, a fair comparison is under roughly the same FLOPS and memory consumption.

The paper is well-written. The method is well-motivated by the multi-objective optimization perspective. Although the presentation of the Hypervolume maximization method (Section 3.2) is not clear, the resulting loss function (Equation 10) is simple, and shares the same form with other previous methods. The hyperparameter \eta is problematic in the new formulation. The authors propose the Nadir Point Adaption to set this parameter.

The authors conduct extensive experiments to compare different methods. The authors emphasize that the performance is improved with more discriminators, but it’s good to contain comparison of the computational cost (FLOPS and memory consumption) at the same time. There are some small questions for the experiments. The reported FID is computed from a pretrained classifier that is specific to the dataset, instead of the commonly used Inception model. I recommend the authors also measure the FID with the Inception model, so that we have a direct comparison with existing reported scores.

Overall, I found that this work is empirical, and I’m not convinced by its experiments about the advantage of multiple-discriminator training, due to lacking of fair computational cost comparison with single-discriminator training.

---

> ### Author Response · Authors · 2018-11-10
> **Response to Reviewer 3**
>
> We thank the reviewer for the suggestions and constructive feedback.
>
> In the following, we quote the reviewer and respectively respond to the specific concern right below.
>
> “Overall, the proposed method is empirical and the authors show its performance by experiments.”
>
> We acknowledge the bulk of evidence for adopting our method is empirical. However, we specifically build upon earlier guarantees introduced by Neyshabur et al. (2017), showing that when approximation along a sufficient number of projections (and discriminators as a consequence) is achieved, the distribution induced by the generator converges to the real data distribution. We thus introduce a more suitable optimization framework to ensure approximation along as many projections as possible, which is not enforced by simply optimizing for the loss average if there is some trade-off along different projections.
>
> “Although the presentation of the Hypervolume maximization  [...] form with other previous methods.”
>
> We apologize for the lack of clarity in this section. Section 3.2 is a brief review of the Hypervolume formal definition for the more general multi-solution case. We tried to make the single-solution case, employed in our work, more clear and intuitive in Section 4.1, and illustrated it with an example in Fig. 1, in which a single solution l has its hypervolume highlighted for a given nadir point \eta. Maximizing the highlighted volume implies minimizing l1 and l2 simultaneously. We added an illustration to Appendix F which might be useful to understand the loss behavior throughout training.
>
> “First, I want to discuss the significance [...] performance improvement. Maybe I’m wrong.”
>
> We agree with the reviewer. The computational complexity of training GANs under a multiple discriminator setting is higher by design in terms of both FLOPS and memory, if compared with single-discriminators settings. However, such setting constitutes not an alternative approach for the recent advances in single-discriminator training, but rather a complementary method which can be used together with other methods.
>
> We would also like to make a few practical remarks regarding the use of multiple discriminators:
>
> 1-While using multiple discriminators may increase the cost of a single training run, the overall cost of training, if one accounts for the several training runs required for hyperparameters search, is reduced. We observed such a behavior in our experiments, as reported in Fig. 5. The reduced variation in training outcomes makes it faster to find a stable training setting when more discriminators are employed. In our point-of-view, multiple-discriminators settings should be employed along with any training scheme of choice, if enough resources are available. As an example, which will be added to the manuscript as soon as we conclude the new experiments, adding discriminators yield the following relative improvement in terms of FID: DCGAN - 55.21%; LSGAN - 57.93% (we are currently running similar experiments on other GANs such as wGAN-GP and hingeGAN).
>
> 2-While the increase in cost in terms of FLOPS and memory is unavoidable, wall-clock time can be made close to single-discriminators cases since training with respect to different discriminators can be implemented in parallel. Extra cost in time introduced by other frameworks such wGAN or SNGAN cannot be recovered.
>
> 3-All our experiments were performed in single-GPU settings, which supports the claim that multiple-discriminators training is practical enough to be employed in several common use cases.
>
> The main conclusions we were able to draw from our experiments is that employing multiple discriminators is a practical approach allowing us to trade extra capacity (and thereby extra computational cost) for higher quality and diversity of generated samples when compared to the single-discriminator equivalent setting, while avoiding mode-collapse and divergence during training for a wider set of hyperparameters.
>
> “From my perspective, a fair comparison [...] FLOPS and memory consumption.”
>
> We understand the concern in terms of fairness of comparison and thank the reviewer for the valuable comment. However, the experiments in the paper were designed to show the effect of adding the extra complexity (in terms of number total parameters) specifically through increasing the number of discriminators (and using random projections+HV loss) in the generated samples. We wanted to show the added cost would translate into performance gain.
>
> “The reported FID is computed from  [...] comparison with existing reported scores.”
>
> We reported in Table 5 in Appendix C the Inception Score and FID with Inception model trained on Imagenet relative to DCGAN. We highlight that our scores are not directly comparable with values reported in other works since we used an upscaled version of CIFAR-10 at 64x64 in order to use the same setting as our main baseline, Neyshabur et al. (2017).

---

> > ### Author Response · Authors · 2018-11-14
> > **Cost-performance analysis and comparison with existing reported scores**
> >
> > We added the suggested computational cost analysis in terms of FLOPS and memory consumption for a complete training step to Appendix C.2. We compared DCGAN, LSGAN, and HingeGAN with their corresponding 24-discriminators versions. We also reported the best FID obtained during training. In summary, these results show that the introduced extra computational cost yields a relevant improvement on the best FID for all cases. We highlight that the increase in performance is solely due to the use of the multiple-discriminators set-up, as all the other aspects were kept unchanged for the same GAN type.
> >
> > Regarding the comparison with other existing results, we hope that Appendix C.4 will address the reviewer’s concerns. We run our method with 24 discriminators using a DCGAN-like generator as described in [1] on CIFAR-10 in its more commonly used version (32x32). We compared the models in terms of FID and Inception Score (using original implementations in TensorFlow) with the results reported in [1] for SNGAN, DCGAN, and WGAN-GP. Furthermore, we implemented our version of SNGAN [1] and, in this case, we also reported the best FID-ResNet obtained during training. This experiment shows that using our approach will improve the performance of a simple DCGAN-like generator to yield FID and Inception Score on-par with the values reported in [1].
> >
> > We believe the added results helped us to strengthen our contribution and we thank once more the reviewer for the thoughtful feedback.
> >
> > [1] Miyato, T., Kataoka, T., Koyama, M., & Yoshida, Y. (2018). Spectral normalization for generative adversarial networks. arXiv preprint arXiv:1802.05957.

---

> > ### Comment · AnonReviewer3 · 2018-12-12
> > **reply**
> >
> > Thank the authors for detailed replies.
> >
> > "if one accounts for the several training runs required for hyperparameters search, is reduced." This is a good point, although it's an empirical (even case-by-case) finding. Figure 5 shows its insensitivity w.r.t. to \delta in your case, how about learning rate in you case? How about learning rate in other cases? In your case, improvement from 8 to 16 is little, but 16 to 24 is significant. Then how many discriminators are enough to obtain this sensitivity? If all these questions have to be answered empirically (through experiments), then the saving in hyperparameters search is discounted...
> >
> > "While the increase in cost in terms of FLOPS and memory is unavoidable" Why is it unavoidable? Can we design an experiment to compare the Inception/FID of single and multiple discriminators, with same FLOP and/or Memory? Or with the same Inception/FID, compare the FLOP and/or Memory? The results in the paper are not enough to answer this controlled experiment question. The added Table 6 cannot, either. By the way, this seems to be a typo.  "This effect was consistently observed considering 4 different well-known approaches, namely DCGAN (Radford et al., 2015), Least-square GAN (LSGAN) (Mao et al., 2017), and HingeGAN (Miyato et al., 2018). "
> >
> > "We wanted to show the added cost would translate into performance gain." Again, if we do not have controlled experiment results, how can we really know whether the performance gain is significant enough so that the added cost is tolerable? "the added cost would translate into performance gain." is the basic requirement for a valid method, but tell nothing that the method is good.
> >
> > Therefore, I prefer to keep my rate. A fair comparison between single and multiple discriminators is my main concern.

---

> > > ### Author Response · Authors · 2018-12-12
> > > **Reply to Reviewer 3**
> > >
> > > We first thank the reviewer for her/his reply.
> > >
> > > The more discriminators one can use the better. This is inline with findings in [1 - Theorem A.2]. We found 24 discriminators to work very well across different datasets and models.
> > >
> > > As we have previously mentioned, the comparison between 1 and 24 discriminators is unfair by design in terms of computational cost. This is exactly the reason why many of our experiments are focused on comparing different multiple-discriminators methods so as to emphasize the importance of our contribution under this setting. The same ”unfairness” is also observed if one compares different generators (DCGAN-like vs. ResNet-based in [2 - Table 2] and [3 - Table 3], for example). This is exactly the point we want to make. If one has the available resources, increasing the cost via adding discriminators is a practical approach to trade the added cost for extra quality/diversity.
> > >
> > > If we understood correctly the experiment suggested by the reviewer, we would have to modify the generator's or discriminator’s architectures to compensate for the extra discriminators. Although very interesting, this experiment would not support any claim we make.  We clarify once more that our focus is on improving the performance of a given generator.
> > >
> > > We empirically support the claim that a given generator will achieve better performance with multiple discriminators when compared to its single-discriminator counterpart. This was consistently observed using generators of various sizes in different datasets.
> > >
> > > For a given generator, using the random projections setting from [1], increasing the number of discriminators will:
> > >
> > > 1-Improve sample diversity
> > > 2-Yield higher sample quality in terms of FID (and Inception score as well)
> > > 3-Make it easier to find a working set of hyperparameters (all default Adam hyperparameters usually yields great improvements w.r.t. single-discriminator)
> > >
> > > By looking at the multiple-discriminators setting through the lens of multi-objective optimization, one can see that optimizing for the average loss will yield solutions at any part of the pareto front, while hypervolume maximization prefers central regions. This is where we believe lies the benefit in using this approach over average loss, since it will enforce the assumptions of [1 - Theorem A.2] (cf. discussion with reviewer 2).
> > >
> > > The reviewer mentioned that “Again, if we do not have controlled experiment results, how can we really know whether the performance gain is significant enough so that the added cost is tolerable?” It is not possible to tell in absolute terms whether the added cost is “tolerable” or not. In our experiments, we were able to train generative models of image data of up to 256x256x3 against 24 discriminators in a single common GPU (NVIDIA GTX 1080ti). Given that, we claim the method is practical, theoretically grounded, and yields relevant performance gains, which we believe is an useful finding for the community to be aware of.
> > >
> > > [1] Neyshabur, Behnam, Srinadh Bhojanapalli, and Ayan Chakrabarti. "Stabilizing GAN training with multiple random projections." arXiv preprint arXiv:1705.07831 (2017).
> > > [2] Miyato, T., Kataoka, T., Koyama, M., & Yoshida, Y. (2018). Spectral normalization for generative adversarial networks. arXiv preprint arXiv:1802.05957.
> > > [3] Gulrajani, I., Ahmed, F., Arjovsky, M., Dumoulin, V., & Courville, A. C. (2017). Improved Training of Wasserstein GANs. In Advances in Neural Information Processing Systems (pp. 5767-5777).

---

> ### Author Response · Authors · 2018-12-01
> **Feedback**
>
> Thank you once more for your time and suggestions. We hope to have addressed your concerns and would appreciate if you take the results included during the rebuttal into consideration when reviewing your score. We are looking forward to hearing back from you and open to discuss any further concern.

---

### Official Review · AnonReviewer1 · 2018-11-05
**A comparison of various weighting approaches to multi-discriminator training**

**Rating:** 6
**Confidence:** 3

**Review:**

Clarity:
The work is a clear introduction/overview of this area of research. The reviewer enjoyed the connections to Multiple-Gradient Descent and clear distinctions/contrasts with previous approaches to weighting the outputs of multiple discriminators. All in all, the paper is quite clear in what its contributions are and how it differs from previous approaches. The details and motivations of the Hypervolume Maximization  (HVM) method (especially as it relates to and interacts with the slack method of picking the nadir point) were a bit harder to follow intuitively given the standalone information in the paper.

Originality:
Adapts a technique to approximate MGD called HVM (Miranda 2016) and applies it to multi-discriminator training in GANs. As far as the reviewer is aware, this is a novel application of HVM to this task and well motivated under the MGD interpretation of the problem.

Significance:
Unclear. This work in isolation appears to present an improvement over prior work in this sub-field, but it is not obvious that the findings in these experiments will continue to be robust in more competitive settings. For instance, the worst performing model on CIFAR10, WGAN-GP (according to the experiments run) WGAN-GP also holds near SOTA Inception scores on CIFAR10 when appropriately tuned. Without any experimental results extending beyond toy datasets like MNIST and CIFAR10 the reviewer is not confident whether fundamental issues with GAN training are being addressed or just artifacts of small scale setups. Closely related previous work (Neyshabur 2017) scaled to 128x128 resolution on a much more difficult dataset - Imagenet Dogs but the authors did not compare in this case.

Quality:
Some concerns about details of experiments (see cons list and significance section for further discussion).

Pros:
+ The work provides a clear overview of previous work on approaches using multiple discriminators.
+ The connections of this line of work to MGD and the re-interpretation of various other approaches in this framework is valuable.
+ The author provides direct comparisons to similar methods, which increases confidence in the results.
+ On the experiments run, the HVM method appears to be an improvement over the two previous approaches of softmax weighting and straightforward averaging for multiple discriminators.


Cons:
- Performance of GANs is highly dependent on both model size and compute expended for a given experiment (see Miyato 2018 for model size and training iterations and Brock 2018 for batch size). Training multiple discriminators (in this paper up to 24) significantly increases compute cost and effective model size. No baselines controlling for the effects of larger models and batch sizes are done.
- The paper lacks experiments beyond toy-ish tasks like MNIST and CIFAR10 and does not do a good job comparing to the broader established literature and contextualizing its results on certain tasks such as CIFAR10 (reporting ratios to a baseline instead of absolute values, for instance). The absolute inception score of the baseline DCGAN needs to be reported to allow for this. Is the Inception Score of the authors DCGAN implementation similar to the 6 to 6.5 reported in the literature?
- Figure 3 is slightly strange in that the x axis is time to best result result instead of just overall wallclock time. Without additional information I can not determine whether it is admissible. Do all models achieve their best FID scores at similar points in training? Why is this not just a visualization of FID score as a function of wallclock time? A method which has lower variance or continues to make progress for longer than methods which begin to diverge would be unfairly represented by the current Figure.

Additional comments:

In section 3.1 Eq 5 appears to be wrong. The loss of the discriminator is presented in a form to be minimized so exponentiating the negative loss in the softmax weighting term as presented will do the opposite of what is desired and assign lower weight to higher loss discriminators.

In Fig 6 FID scores computed on a set of 10K samples are shown. The authors appear to draw the line for the FID score of real data at 0. But since it is being estimated with only 10K samples there will be sampling error resulting in non-zero FID score. The authors should update this figure to show the box-plot for FID scores computed on random draws of 10K real samples. I have only worked with FID on Imagenet where FID scores for random batches of 10K samples are much higher than 0. I admit there is some chance the value is extremely low on CIFAR10 to make this point irrelevant, however.

---

> ### Author Response · Authors · 2018-11-10
> **Response to Reviewer 1**
>
> We thank the reviewer for the thoughtful comments and feedback.
>
> We quote the reviewer and address the respective comments below.
>
> “The details and motivations of the Hypervolume Maximization (HVM) method [...]”
> We thank the reviewer for pointing this out. Given the limitation in space in the main text, we added two figures to Appendix G in the hope this will make more clear how the hypervolume interacts with nadir point coordinates values.
>
> “Significance:
> Unclear. This work in isolation appears to present an improvement over prior work [...]”
>
> We included samples of generators trained on CelebA at 128x128 in Appendix D for different numbers of discriminators. Architectures correspond to the ones used for the 64x64 case, with 1 extra conv. layer in both models.
>
> In order to further emphasize the significance of our contributions, we would like to highlight the following points.
> 1-In the coverage evaluation performed on top of stacked MNIST, results reported were computed using 10k generated samples while results reported in previous literature are computed employing a sample of size 26k. Both scenarios are now included in the last uploaded version, and after repeating the evaluations using 26k images we were able to cover the maximum number of 1000 modes with 16 and 24 discriminators, and 776.8+-6.4 modes with 8 discriminators.
>
> 2-Regarding WGAN-GP, our implementation obtained worse FID-ResNet (trained on CIFAR-10)  than DCGAN. On the other hand, Inception Score and FID with Inception model were both better with WGAN-GP, as reported in literature.
>
> Cons
> 1-“Performance of GANs is highly dependent on both model size and compute [...]”
>
> We focused in going from a single- into the multiple-discriminators case, while keeping the generator architecture and training setting unchanged. This is done to isolate the effect of the added discriminators. We acknowledge the fact that different architectures will benefit differently from the added discriminators, however we observed similar effects in all cases considered within this work.
>
> We further highlight the multiple-discriminator setting is not an alternative to other training schemes for GANs, but rather a complementary training strategy that can (and should, in our view) be used together with other methods. As such, our experiments are intended to (i)-show the effect given by the addition of discriminators, and (ii)-show that hypervolume maximization provides an effective “policy” to assign importance to different discriminators.
>
> 2-“The paper lacks experiments beyond toy-ish tasks [...]”
>
> We decided to report Inception Scores as a ratio with respect to DCGAN since they are not directly comparable to most of the values reported in literature. In order to keep a consistent comparison with our main baseline, Neyshabur et al. (2017), we decided to employ exactly the same architecture, which was designed for inputs of size 64x64. We thus upscaled CIFAR-10 and this changes the scores range. Inception Score obtained by DCGAN in the 64x64 rescaled version of CIFAR-10 was 4.0697+-0.0861 (10 runs with 10k samples). Moreover, aiming to better contextualize our contribution with other approaches, we are running experiments with the 32x32 version of CIFAR-10.
>
> 3-“Figure 3 is slightly strange in that the x axis is time to best result instead of just overall wallclock time. [...]”
>
> Following the reviewer’s suggestion, we added the suggested plot to Appendix F.
>
> All the models are trained with a fixed budget in terms of iterations (93800, corresponding to 100 epochs with a batch size of 64). Our goal was indeed to emphasize a trade-off between faster convergence to a sub-optimal FID vs. later convergence to a better value. AVG and GMAN were not able to further improve FID after a few training iterations. On the other hand, HV was able to further improve the achieved best FID and MGD could take even more advantage of the available training budget, as it was able to decrease the FID almost until the end of training.
>
> Additional comments:
> 1-“In section 3.1 Eq 5 appears to be wrong. [...]”
> Indeed the minus signs on the betas are typos on the definition of alpha_k’s (we also double-checked our implementation and it is correct). We fixed this on the updated version of the manuscript.
>
> 2-“In Fig 6 FID scores computed on a set of 10K samples are shown. [...]”
>
> We agree and to further investigate this we compared FID values obtained for real data using three different architectures, namely Inception-V3, ResNet18 and VGG-16. The model to calculate FID using Inception-V3 was trained on Imagenet. More specifically, we first compute the statistics of the training partition of CIFAR-10 and then compute the FID for the test set. Obtained values were 3.1796, 0.0319, and 0.0255, respectively, with very small variation. As pointed out by the reviewer, since CIFAR-10 has only 10 classes, using 10k real samples to calculate FID should have a smaller sampling error in comparison with Imagenet.

---

> > ### Author Response · Authors · 2018-11-14
> > **Comparing to the broader established literature**
> >
> > We want to let the reviewer know that suggested experiments were included in the newest version of our manuscript. We point the reviewer to:
> >
> > Appendix D.2 for generated CelebA samples at 128x128 under varying number of discriminators.
> > Appendix E for generated Cats samples at 256x256. Notice that we used only training 1740 samples.
> > Appendix C.4 in which we included results in the original CIFAR-10 for a more clear comparison with other methods. We thus show that adding multiple discriminators with HVM training will shift performance of a vanilla DCGAN-like generator to scores inline with [1].
> >
> > We hope to have addressed the reviewer’s concerns.
> >
> > [1] Miyato, T., Kataoka, T., Koyama, M., & Yoshida, Y. (2018). Spectral normalization for generative adversarial networks. arXiv preprint arXiv:1802.05957.

---

> ### Author Response · Authors · 2018-12-01
> **Feedback**
>
> Thank you once more for your time and suggestions. We hope to have addressed your concerns and would appreciate if you take the results included during the rebuttal into consideration when reviewing your score. We are looking forward to hearing back from you and open to discuss any further concern.

---

### Author Response · Authors · 2018-11-10
**Summary of modifications**

We thank the reviewers for their time reading our paper and for providing useful feedback. We summarize the main corresponding modifications in the updated version of the manuscript in the following:

- Updated stacked MNIST results with extra evaluation for test sets of different sizes. Evaluations with 10000 and 26000 images (as usually reported) are now shown in Table 1.
- Added to Appendix G a plot of minimum FID vs. wall-clock time for MNIST experiments in order to aid the understanding of Figure 3, as suggested by Reviewer 1.
- Illustration added to Appendix F to make Section 4.1 easier to follow.
- Added samples obtained on CelebA 128x128 with 6, 8, and 10 discriminators to Appendix D.2 in order to show that the proposed method scales-up to higher resolution datasets.

---

### Author Response · Authors · 2018-11-14
**Further updates**

In this post we describe a number of new experiments that we performed in response to the reviewers’ questions. We believe that these results strengthen the relevance of the discussed framework and thank the reviewers; their helpful suggestions were very useful in improving our work.

1-As suggested by Reviewer 1, we ran experiments on CIFAR-10 at its standard resolution (32x32) for a clear comparison with previous approaches. Results, now shown in Appendix C.4 - Table 7 include a comparison with SNGAN [1], in which we show adding multiple-discriminators with HVM in a DCGAN-like setting yields relevant improvements in both FID and Inception Score.

2-Following suggestion of Reviewer 2, we included Table 6 in Appendix C.2, in which we compare single- vs. multiple-discriminators settings of 3 GANs in terms of FID and computational cost. Results support the claim that the added cost yields higher quality samples, which was consistently observed across the different settings.

3-To further address Reviewer’s 1 concern as to whether our method scales-up to higher resolution datasets, we added generated images of size 256x256 obtained by a generator trained with 24 discriminators on the Cats dataset, containing only 1740 training examples, as similarly done in [2]. Notice that adding the multiple discriminators setting allowed us to successfully train the same generator that was shown in [2] to not be able to yield samples that look natural (see Figures 4 and 5 in [2]). Generated samples are now presented in Appendix E.

We highlight that all experiments performed within this work were executed in single GPU hardware, which indicates the multiple discriminator setting is a practical approach.

We believe that our new, stronger results address issues brought up by the reviewers (also cf. individual reviewer responses) and hope that the reviewers will kindly consider our improvements for their final evaluation.

[1] Miyato, T., Kataoka, T., Koyama, M., & Yoshida, Y. (2018). Spectral normalization for generative adversarial networks. arXiv preprint arXiv:1802.05957.
[2] Jolicoeur-Martineau, Alexia. "The relativistic discriminator: a key element missing from standard GAN." arXiv preprint arXiv:1807.00734 (2018).

---

### Meta-Review · Area_Chair1 · 2018-12-15
**A well written paper on multi-discriminator GAN training, but just below the bar in empirical results**

**Confidence:** 5
**Recommendation:** Reject

**Metareview:**

The reviewers found that paper is well written, clear and that the authors did a good job placing the work in the relevant literature.  The proposed method for using multiple discriminators in a multi-objective setting to train GANs seems interesting and compelling.  However, all the reviewers found the paper to be on the borderline.  The main concern was the significance of the work in the context of existing literature.  Specifically, the reviewers did not find the experimental results significant enough to be convinced that this work presents a major advance in GAN training.